# Application of Multi-Slice Computed Tomography for the Preoperative Diagnosis and Classification of Pulmonary Cystic Echinococcosis

**DOI:** 10.3390/pathogens10030353

**Published:** 2021-03-16

**Authors:** Lizhong Wu, Longlong Mu, Mingjue Si, Jie Xu, Guojie Ciren, Lingling Cai

**Affiliations:** 1Department of Radiology, People’s Hospital of Shigatse City, Shigatse 857000, China; woolley@shsmu.edu.cn (L.W.); 13658926368@139.com (P.); mulonglong@126.com (L.M.); 2Department of Radiology, Shanghai Ninth People’s Hospital, Shanghai JiaoTong University School of Medicine, Shanghai 201999, China; smjsh@outlook.com; 3Department of Infectious Disease, Shanghai Ninth People’s Hospital, Shanghai JiaoTong University School of Medicine, Shanghai 201999, China; dr.xu@aliyun.com

**Keywords:** pulmonary cystic echinococcosis, multi-slice computed tomography, preoperative diagnosis, CT-image derived classification

## Abstract

Pulmonary cystic echinococcosis remains a serious threat to public health. A standardized, imaging-based classification method for pulmonary echinococcosis has not yet been developed despite the existence of a standardized ultrasound classification method and treatment plan for hepatic cystic echinococcosis. Chest computed tomography (CT) images from 34 cases of pulmonary cystic echinococcosis with 46 lesions were used for classification based on the World Health Organization (WHO) standardized ultrasound classification of hepatic cystic echinococcosis. CT findings were compared with intraoperative observations and postoperative pathological results to assess accuracy. Pulmonary cystic echinococcosis was common in women (14/34, 41.2%) and children (14/34, 41.2%) with a single cyst (28/46, 60.9%). Most lesions were classified as cystic echinococcosis 1(CE1, 19/46) or cystic echinococcosis 3(CE3, 21/46). Blood leukocytosis was mostly observed in CE3 lesions (100%, 9/9) (*p* < 0.05). The preoperative CT diagnosis of pulmonary cystic echinococcosis had an accuracy rate of 100%. The preoperative CT typing, and postoperative pathological typing had a coincidence rate of 97.8% (45/46). Our study provided a classification method based on CT imaging for pulmonary cystic echinococcosis that can be used during pre-surgical planning to reduce patient’s postoperative complications and mortality.

## 1. Introduction

Echinococcosis is a zoonotic parasitic disease with a worldwide distribution. It is commonly known as hydatidosis or hydatid disease [1]. This disease is caused by the Echinococcus larvae, which has been reported to parasitize humans, cattle, and sheep. Human cystic echinococcosis caused by Echinococcus granulosus remains a serious public health problem, endangering human health in many parts of the world. With a population infection rate of 45.1 per 100,000 people, China has a high incidence of echinococcosis [2]. Echinococcosis cases are mainly concentrated in arid pastoral areas in the west of China, such as in the Tibet Autonomous Region.

The most common site of cystic echinococcosis is the liver and is mainly assessed with ultrasonography [1]. The World Health Organization Informal Working Group on Echinococcosis (WHO-IWGE) classified hepatic cystic echinococcosis into six types based on the Gharbi ultrasound classification method [3,4]. Currently, the WHO-IWGE classification standard is the most widely used standard and is used to determine treatment plans for hepatic cystic echinococcosis [5]. The lung is the second most common site of adult cystic echinococcosis, accounting for 10–40% of all cases [6,7]. Previous studies have shown that surgical treatment is currently the safest and most effective method for pulmonary cystic echinococcosis (PCE) [7,8]. The Expert consensus clearly states that all pulmonary cysts should be operated on [9]. The success of surgical treatment is reliant upon performing a comprehensive lesion assessment before the operation and determining an assessment-based treatment plan [8]. Since multi-slice computed tomography (MSCT) has high density and spatial resolution, it can provide detailed imaging information that is useful for evaluating PCE lesions. Thus, MSCT is the preferred imaging method for PCE.

For this study, we retrospectively examined 34 PCE cases with chest CT images and surgical pathology results from the Shigatse region of Tibet. Each lesion was classified using preoperative CT data based on the WHO-IWGE ultrasound classification criteria for hepatic cystic echinococcosis. Following this, the accuracy of preoperative CT classification and its value for formulating the surgical plan were verified through comparisons with intraoperative findings and postoperative pathology.

## 2. Results

### 2.1. Patient Demographics

Patient demographics are summarized in Table 1. Of the 34 patients, women and children constituted most cases, both accounting for 41.2% (14/34) each of all cases. In contrast, men only accounted for 17.6% (6/34). The average patient age was 31.1 years (range, 4–62 years), and the average ages of men, women, and children were 46 (19–62), 46.8 (20–62), and 9.1 years (7–13 years), respectively. There was no statistically significant difference in age between male and female patients (*p* = 0.207).

Of the 34 patients, 11 patients had CE1 lesions (32.4%, 11/34), 3 patients had CE2 lesions (8.8%, 3/34), 14 patients had CE3 lesions (41.2%, 14/34), 4 patients had lesions of CE1 + CE3(11.8%, 4/34), 1 patient had lesions of CE3 + CE5(2.9%, 1/34), 1 patient had lesions of CE1 + CE4(2.9%, 1/34). No patient had single lesions of neither CE4 nor CE5.

Routine blood examination showed that most cases with leukocytosis had CE3 lesions (100%, 9/9). In contrast, cases with other lesion types had basically normal leukocyte counts. This difference was statistically significant (*p* = 0.026). However, there was no statistically significant difference in eosinophil elevation between different CT type groups (*p* = 0.198) (Table 2).

### 2.2. CT Image Analysis and Classification

#### 2.2.1. CT Parameters

CT provided detailed information on the relationships of the cysts with adjacent structures and the cysts’ number, size, and location (Table 3).

#### 2.2.2. CT Image Analysis

The preoperative CT findings of the 46 PCE lesions were classified according to the WHO ultrasound classification of hepatic cystic echinococcosis. There were no cases of CL lesions found. These lesions demonstrate unclear development on CT and have a thin cyst wall and contain fluid similar to water. There were 19 cases with CE1 lesions. These lesions have a round or oval shape with a thin, well-defined, and smooth cyst wall of relatively high density, containing water-like fluid, and there is potentially a gap between the inner and outer cyst walls, resembling a “double-wall” (Figure 1a). CE1-type cysts and cyst walls were not enhanced by the contrast agent. There were two cases of CE2 lesions. These lesions are reported as two or more relatively small cysts, which have a representative “cyst-in-cyst” appearance (Figure 1b). There were 21 cases of CE3 lesions, of which 19 cases were CE3a and 2 cases were CE3b. These lesions have a complicated appearance following the rupture of hydatid cysts. Gas had entered between the internal and external cysts in two cases, resembling a “crescent” shape (Figure 2c). In another case, the inner and outer cysts had ruptured, and the internal cyst attached itself to the wall of the external cyst without collapsing, which resulted in a “double-crescent” shape (Figure 2d,e). In yet another case, the cyst wall had collapsed, curled, and wrinkled and was floating in cystic fluid, taking the appearance of a “ribbon” shape (Figure 2f). In 17 cases, the cyst wall laceration connected with the bronchi, and the cyst contents had been discharged. Additionally, gas had entered the cystic cavity, resulting in either a “floating lotus” appearance when the cavity contained more cystic fluid (7 cases, Figure 2g) or a “stone-emerging-from-water” appearance when the cavity contained less fluid (10 cases, Figure 2h). Coronal and sagittal MSCT MPR (multi-planner reformatted) images showed that these lesions were strongly connected to the bronchi. There was one case of a CE4 lesion that had a solid appearance that was not enhanced with contrast agent on CT (Figure 3). One case had adhesion of a cystic echinococcosis lesion in the lower lobe of the right lung, which penetrated through the diaphragm (Figure 4). Additionally, there were three cases of CE5 lesions. These lesions are characterized by thick, calcified cystic walls and visible, irregular strip-like and arc-shaped calcifications (Figure 1e).

Maximum lesion diameters were measured on transverse CT images, with small (<5 cm) lesions accounting for 28.3% (13/46), medium (5–10 cm) lesions for 47.8% (22/46), and large (>10 cm) lesions for 23.9% (11/46) of all lesions.

In addition, pericystic pulmonary exudation and consolidation were demonstrated in 85.3% (29/34) of the patients. Atelectasis was observed in 64.7% (22/34) of the patients, whereas pleural effusion and pneumothorax were noted in 52.9 (18/34) and 2.9% (1/34) of the patients, respectively.

### 2.3. Surgical Methods and Pathological Results

Of the 19 cases with CE1 lesions, five cases had small lesions (<5 cm). Of these cases, two underwent complete lesion resection, one underwent wedge resection of the lung segments, and two underwent lower lobe resection. The latter cases had multiple lesions (CE1 + CE3 and CE1 + CE4, respectively) located in the lower lobe of the lung. Additionally, there were 14 cases with medium and large lesions (≥5 cm). Of these cases, seven underwent complete resection, two underwent partial resection, three underwent wedge or segmental resection of the lung, one underwent internal cyst resection, and one underwent a lobectomy.

Two cases had CE2 lesions that were medium-sized (5–10 cm) and underwent complete resection.

Of the 21 cases with CE3 lesions, six cases had small lesions (<5 cm). Additionally, of these cases with small lesions, two underwent complete lesion resection, one underwent wedge or segmental resection of the lung, one underwent internal cyst resection, one underwent lobectomy, and one underwent complete lesion resection combined with diaphragmatic echinococcosis resection and muscle repair. Furthermore, there were 15 cases with medium and large lesions (≥5 cm). Of these cases, five underwent complete lesion resection, two underwent wedge or segmental lung resection, and two underwent internal cyst resection. The remaining six underwent a lobectomy.

One case had a CE4 lesion with mediastinal lymph node enlargement, and threes cases had CE5 lesions. These lesions were small (<5 cm) and underwent complete lesion resection.

The CT-based preoperative diagnosis of PCE had an accuracy rate of 100%. Preoperative CT typing and postoperative pathological typing had a coincidence rate of 97.8% (45/46). Only one CE2-type case was incorrectly classified as a CE1-type case (Table 4).

## 3. Discussion

Cystic echinococcosis is the most common type of echinococcosis worldwide [1]. In our study, the incidence rates in women (41.2%, 14/34) and children (41.2%, 14/34) were significantly higher than that in men (17.6%, 6/34), which may be related to the lives, eating habits, and production methods of local Tibetans. Moreover, women mostly performed the work in these pastoral areas. Therefore, the likelihood of direct contact with hosts, such as cattle and sheep, were significantly higher for women than men. The high rate of infection in children may be related to their outdoor play behavior, which increases the chance of contact with potentially contaminated soil and could be compounded by frequent hand-to-mouth contact [10].

About 60–75% of PCE patients are asymptomatic in the early stage of the disease [10]. PCE lesions grow slowly and are often latent for several years, leading to difficulties in the early diagnosis of echinococcosis. In this study, most of the patients were detected by chest X-ray or CT examination performed for a concurrent upper respiratory tract infection or other reasons. Ozyurtkan et al. [6] found that elevated leukocytes are more common in cases with hydatid cyst rupture. In the present study, 100% (9/9) of cases with leukocytosis had cyst rupture, which is consistent with previous findings [6,11] and indicates the presence of secondary infection of the CE3 lesion through cyst wall rupture. Parallel infections might have influenced measured blood parameters such as leucocyte count. In this study, we did not consider this bias and therefore correlation between increased leukocytes and stages of the cysts still needs to be verified. Additionally, in this study, there was no correlation between eosinophilia and cyst rupture. Eosinophils are suggestive of parasitic infection. The sensitivity and specificity of the current immunodiagnostic tests for echinococcosis are low [12,13], and their diagnostic value is limited. Therefore, we opted not to perform these tests on our patients. Ultrasonography is the first choice for diagnosing hepatic echinococcosis [14,15]; however, when it is used for the lungs, intrapulmonary gas as well as rib and scapula blockage can easily interfere with its results. Chest X-ray can overcome some of the limitations of ultrasonography; however, PCE presents as a quasi-circular mass shadow with a clear boundary on X-ray. Moreover, it cannot show the complete shapes and details of the lesions [8], making it difficult to distinguish from other lung and mediastinal masses. The original image data of MSCT can undergo multi-azimuth and -plane reconstruction to obtain 2D and 3D images of any orientation. MSCT can accurately locate PCE lesions, classify their morphology, evaluate the condition of the surrounding lung tissue, and determine whether complicated infection or atelectasis is present. Moreover, MSCT provides detailed and reliable information for the formulation of a surgical plan. For example, when an internal cyst has ruptured, MSCT can clearly show the communication between the crevasse and adjacent bronchi, the relationship between the crevasse and the surrounding large blood vessels and mediastinum, and the presence of diaphragm involvement. Therefore, MSCT has significant advantages for diagnosing pulmonary echinococcosis, making it the first choice for diagnosis and necessary before surgery. In our study, preoperative MPR reconstruction showed one case with adhesion of a cystic echinococcosis lesion in the lower lobe of the right lung. Therefore, a surgical plan was established immediately before the operation to perform pulmonary and diaphragmatic resection and repair.

We found that PCE has certain distinctive characteristics. First, most cases of PCE only involve single cysts (28/46, 60.9%) and multiple cysts are rare. Moreover, pulmonary echinococcosis lesions are often large, averaging 7.9 cm. This finding may be due to the presence of loose lung tissue, rich blood circulation, adequate nutrition, and negative pressure in the thoracic cavity, which results in the faster growth and larger size of pulmonary echinococcosis lesions. Second, PCE was observed to occur in the lower lobes of the lungs (60.9%, 28/46), which is consistent with previous studies [16]. PCE is mostly distributed on the periphery of the lung or the surface of the interlobar fissure. This may be related to the tendency of oncospheres to move with the blood flow to the lung and stay at the distal capillary end [17]. Third, CE3 lesions may have many complicated manifestations due to internal cyst rupture, which resemble “crescent”, “ribbon”, “floating lotus”m “stone-emerging-from-water”, or “double-crescent” shapes. Imaging physicians should fully understand these characteristic CT manifestations indicating rupture. Hydatid cyst fluid is an antigen [16] and is known to cause severe allergic reactions or even be fatal following cyst rupture. The quick diagnosis and evaluation of CE3 lesions are critical for determining the appropriate surgical treatment. Fourth, the CE4 lesion is cystic and solid, and the solid component is an *Echinococcus* cyst. In the CE4 lesion, the cystic fluid is absorbed and the cyst wall folds and shrinks. Subsequent necrosis, dissolution, and denaturation occur shortly after. As a result, there is no enhancement on CT, which can help distinguish this lesion from lung abscesses and malignant tumors. Fifth, the CE5 lesion is rare in pulmonary echinococcosis, accounting for only 6.5% (3/46) of all cases. Such cases have fine-line calcifications and small cyst walls. This differs from hepatic echinococcosis, which is characterized by irregular strip-like or eggshell cyst wall calcification, as well as complete calcification.

The present study found that CT exhibited a 100% preoperative diagnostic accuracy rate for pulmonary echinococcosis. The classification method for liver echinococcosis by ultrasonography was applied to the classification of lung echinococcosis by CT, and the coincidence rate of this approach with postoperative pathological results reached very high levels of up to 97.8% (45/46). Only one CE2 case was mistakenly classified as a CE1 case. In this case, the cyst wall was very thin and a low-dose CT scan was performed, which prevented the cyst wall from being clearly displayed. Therefore, MSCT is a feasible and accurate imaging method for classifying PCE into the six lesion types defined by the WHO-IWGE.

Once PCE is definitively diagnosed, surgical treatment should be immediately considered if the patient has suitable cardiopulmonary function. Currently, surgical treatment is the most effective and safest treatment method [18,19]. The goals of surgery include removal of as much of the PCE lesion as possible, retention of the maximum amount of functional lung tissue, and absolute avoidance of cyst fluid overflow during the operation [7,12]. The specific operation plan is mainly based on lesion location, size, and classification as well as the presence of lesion complications found on preoperative CT [17]. Our study showed that the complete resection rate of CE1 and CE2 lesions without complications was 71.4% (15/21). This resection rate applied to wedge resection of pulmonary hydatid cysts with diameters <2 cm that were near the lung surface as well as to segmental resection of lesions located in the whole lung segment. Partial or simple excision of the internal cyst was applied to 14.3% (3/21) of lesions that were complicated by severe infection and adjacent lung tissue adhesion. For CE3 lesions, the complete or segmental resection rate was 53.4% (11/21). Moreover, 14.3% (3/21) of these lesions had large volumes and adhered closely to the surrounding lung tissue, which necessitated internal cyst resection. A pulmonary lobectomy was performed for 33.3% (7/21) of large lesions with a secondary severe infection. In our study, CE4 and CE5 lesions were small and underwent complete resection. A previous study has suggested that CE4 and CE5 lesions are inactive and generally have few complications [15]. According to the treatment results of our study, we recommend that patients with CE1, CE2, CE4, and CE5 type mainly undergo hydatid cyst resection and patients with CE3 type undergo pulmonary segmental or lobectomy according to the scope of imaging and whether there are complications. Patients with CE3 who have difficulty in resection are advised to undergo internal capsule removal.

The limitations of this study are its small sample size, limited follow-up time, and bias. All of the selected cases were surgical cases, and data from patients with contraindications to surgery and those who refused surgery were not analyzed. Moreover, long-term follow-up for recurrence after discharge was not performed. In the report from Stojkovic et al., MRI reproduces the ultrasound-defined features of CE better than CT [20]. The CT typing accuracy observed requires further verification through future, prospective studies with larger sample sizes.

In conclusion, we demonstrated that the combination of the WHO-IWGE standardized ultrasound classification method for hepatic cystic echinococcosis with MSCT of the lung provided a high classification accuracy for pulmonary echinococcosis. This approach may serve as a reliable and preferable imaging method for the preoperative evaluation of pulmonary echinococcosis, which can allow patients to receive effective and efficient treatment and effectively reduce surgical complications and patient mortality.

## 4. Materials and Methods

This study was approved by the institutional review board at our hospital. Given the retrospective nature of the study, the requirement for informed consent was waived.

### 4.1. Patient Enrolment

A total of 34 patients diagnosed with PCE and treated at our hospital in Shigatse city Tibet from 2016 to 2018 were all included in this study. Of them, 28 patients had one pulmonary lesion, 3 patients had two pulmonary lesions, 2 patients suffered from three pulmonary lesions, and 1 patient had 6 pulmonary lesions. Patients’ medical history and the blood laboratory reports were retrieved from medical records. All patients underwent plain MSCT scans before the operation, and five patients had additional contrast-enhanced CT. The study was approved by the Institutional Review Board of the People’s Hospital of Shigatse City in Tibet, China.

### 4.2. CT Scanning and Imaging

Patients were scanned either with a 16-detector row CT scanner (Aquilion 16; Toshiba Medical, Tokyo, Japan) or a 128-slice CT scanner (uCT 760; United Imaging, Shanghai, China). The acquisition parameters of the 16-detector row CT scanner were set at 120 kVp of tube voltage, 160 mA maximum automatic tube current modulation, 0.5 s gantry rotation speed, and 16 × 0.5 mm beam collimation. Additionally, the acquisition parameters of the 128-slice CT scanner were set at 120 kV tube voltage, 446 mA maximum uDose^®^ Intelligent Tube Current Modulation, 64 × 0.5 mm collimation, and 1.5 mm at 1-mm increments. The raw data were reconstructed to 0.5 mm thickness at 0.5 mm intervals using a high-spatial-frequency algorithm for lung parenchyma. Multiplanar reconstructions were performed for the morphological features of lesions and adjacent bronchi. Dynamic contrast-enhanced CT imaging was performed following intravenous injection of 40–80 mL of contrast medium (Ioversol, 300 mg iodine/mL; Hengrui, Lianyungang, China) at a rate of 3 mL/s with a power injector.

### 4.3. Diagnostic Criteria and Lesion Classification

The patient was diagnosed with PCE if he/she met the following diagnostic criteria: (1) he/she had a history of dog contact in an epidemic area where dogs were used to take care of cattle or sheep; (2) CT scan findings showed a single or multiple liquid-filled low-density lesions with a round or oval shape in his/her lungs [21]. Ruptured lesions presented with “crescent”, “ribbon”, “floating lotus”, or “stone-emerging-from-water” shapes, which could be accompanied by similar lesions in the liver; (3) two experienced, independent radiologists reached a consensus with their interpretations.

By referring to the definition of the WHO-IWGE ultrasound classification for hepatic cystic echinococcosis, we classified the lung lesions as follows: CL for cystic lesion, CE1 for single-cyst, CE2 for multi-cysts, CE3 for internal cyst rupture, CE4 for consolidation, and CE5 for calcified lesion. Based on the maximum lesion diameter on transverse CT images, we also stratified the lesions as small for <5 cm, medium for 5–10 cm, or large for >10 cm.

### 4.4. Surgery and Pathology

The surgeries included thoracotomy and thoracoscopic surgery. Per the preoperative CT images and intraoperative findings, hydatid or internal cyst excision, wedge or segmental resection of the lung, or lobectomy could be adopted. All surgical specimens were sent to the pathological department for histology. The pathological diagnosis of PCE was based on the gross observation of milky-white [21], sheet-like hydatid or internal cysts as well as the microscopic observation of internal cyst tissue including chitin and an inner germinal layer.

### 4.5. Statistical Analysis

SPSS Statistics V21.0 software was used for statistical analyses (IBM, Armonk, NY, USA). The data were presented as mean ± standard deviation or the number of cases. The χ2 test was used for inter-group percentage comparisons, and a Fisher’s exact probability test was used when the theoretical frequency was <5. Continuous data were compared using the independent sample *t*-test. *p* < 0.05 was considered statistically significant.

## Figures and Tables

**Figure 1 pathogens-10-00353-f001:**
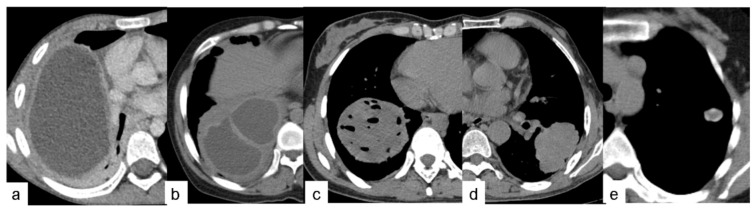
For the 46 PCE lesions included in this study, thin-section CT findings were classified according to the WHO ultrasound classification of hepatic cystic echinococcosis: (**a**) Type of CE1 with single-cyst lesions. (**b**) Type of CE2 with multi-cyst lesions. (**c**) Type of CE3a with internal cyst rupture lesions. (**d**) Type of CE4 with consolidation lesions. (**e**) Type of CE5 with calcified lesions.

**Figure 2 pathogens-10-00353-f002:**
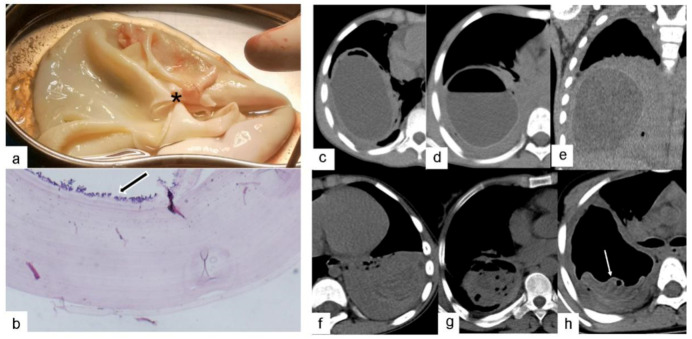
Five cases of CE3a lesions had a complicated appearance following the rupture of hydatid cysts: (**a**) Gross specimen reveals a rupture hole (asterisk) in a whitish-yellow or white gelatinous membrane. (**b**) Low-magnification photomicrograph of a histologic specimen (Hematoxylin-Eosin, 100×) shows a cyst wall infected with *Echinococcus*
*granulosus*. The microscopic section of the endocyst shows a laminated membrane lined on its inner surface by the nucleated germinal layer (black arrows), which produces brood capsules and protoscolex. (**c**) Axial CT image shows a “crescent” shape. (**d**) and (**e**) Axial and coronal CT images exhibit a “double-crescent” shape. (**f**) Axial CT image shows a “ribbon” shape. (**g**) Axial CT image shows a “floating lotus” shape. (**h**) Axial CT image shows a “stone-emerging-from-water” shape (white arrow).

**Figure 3 pathogens-10-00353-f003:**
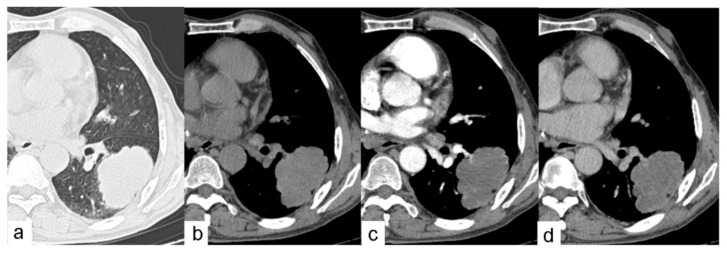
One case of dynamic contrast-enhanced CT scan of the chest in a 62-year-old man. (**a**) Axial CT image shows a PCE lesion in the left lower lobe of the lung-on-lung window. (**b**) Axial non-contrast CT image shows the same lesion on the mediastinal window with a CT value of 32HU. This lesion shows no contrast enhancement on (**c**) arterial phase and (**d**) venous phase.

**Figure 4 pathogens-10-00353-f004:**
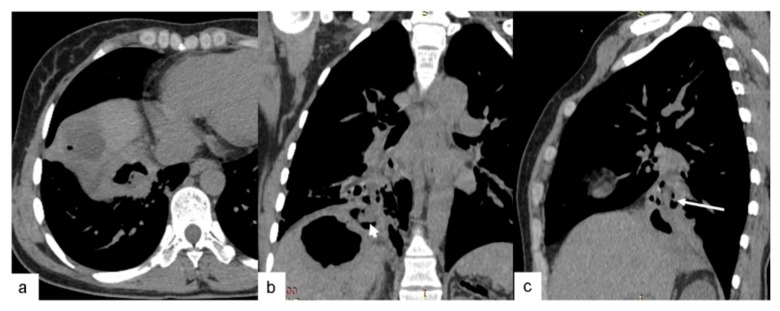
One case of CE3a lesion in the lower lobe of the right lung combined with lesion in the liver in a 35-year-old woman. (**a**) Axial CT image shows a “stone-emerging-from-water” shape when the cyst was ruptured. (**b**,**c**) Coronal and sagittal reformatted CT images demonstrate a hepatic hydatid cyst (small arrow) invading the thoracic cavity throughout the diaphragm (arrow).

**Table 1 pathogens-10-00353-t001:** Patient demographics.

Parameter	No. of Patients (n = 34)
Age (years)	
Mean ± standard deviation (Median, range)	31.1 ± 21.3 (32.5, 4–62)
Sex	
Adult	20
Male	6
Female	14
Children	14
Male	8
Female	6
Nationality	
Tibetan	34 (100%)
**Occupation**	
Herder/Farmer	20
Student	14
Clinical symptoms	
Cough	26
Expectoration	24
Chest pain	25
Dyspnea	9
Hemoptysis	3
Combined with liver echinococcosis	10
Blood routine	
Increased leukocyte count	9
Increased eosinophil count	10
C-reactive protein	21

**Table 2 pathogens-10-00353-t002:** Correlation between increased leukocytes and eosinophils and echinococcosis types.

	Leukocytes (n = 34)			Eosinophils (n = 34)		
	Increased	Normal	*p*	Increased	Normal	*p*
CE1	0	11	0.026	2	9	0.198
CE2	0	3		2	1	
CE3	7	7		3	11	
CE4	0	0		0	0	
CE5	0	0		0	0	
CE1 + CE3	2	2		2	2	
CE3 + CE5	0	1		1	0	
CE1 + CE4	0	1		0	1	
Total	9	25		10	24	

**Table 3 pathogens-10-00353-t003:** CT Results.

Parameter	No. of Lesions (n = 46)
Right	19
Upper lobe	4
Middle lobe	4
Inferior lobe	11
Left	27
Upper lobe	10
Inferior lobe	17
Observation	
Observation size (mm)	
Mean ± standard deviation (Median, Range)	78.70 ± 34.20 (79, 19–173)
Stratified according to size	
Small (<5 cm)	13
Middle (5~10 cm)	22
Large (>10 cm)	11

**Table 4 pathogens-10-00353-t004:** Comparison of preoperative CT typing and postoperative pathological typing.

	Preoperative CT Typing	Postoperative Pathological Typing
CL (cystic)	0	0
CE1 (single-cyst)	20	19
CE2 (multi-cyst)	1	2
CE3 (internal cyst rupture)	21	21
CE4 (consolidation)	1	1
CE5 (calcified)	3	3

## Data Availability

The datasets generated during and/or analyzed during the current study are available from the corresponding author on reasonable request.

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
