# Peer review of "Application of Multi-Slice Computed Tomography for the Preoperative Diagnosis and Classification of Pulmonary Cystic Echinococcosis"

_pathogens, 2021, doi:10.3390/pathogens10030353_

Round 1

Reviewer 1 Report

The Au. describe in a small cohort of patients different stages of CE in the lungs with the aim to translate the stadiation of liver cysts to those in the lungs.

The Au. should read this article before re-submitting the manuscript:

Plos NTD, 2012, Stojkovic et al. Diagnosis and staging of cystic echinococcosis: how do CT and MRI perform in comparison to ultrasound?

The sentence about serology is not correct, in many cases serology may help in the diagnosis, the Au. do not cite any paper to support their position.

Any way, a serological check of the evaluated patients would have improved the quality of the study.

The Au. cite the WHO-IGE, but they are not aware of the Expert consensus which clearly states that all pulmonary cysts should be operated. According to WHO-ICE The stage CE3 is divided in C3a and C3b, but these stages are not considered by the Au.

An informtion which might be added is the presence or not of cysts in other organs, different from the lungs.

Author Response

Dear editor,

Thank you very much for your E-mail regarding our manuscript entitled “Application of multi-slice computed tomography for the preoperative diagnosis and classification of pulmonary cystic echinococcosis”. First of all, we thank editor and reviewers for constructive and helpful comments on our manuscript. We also appreciate the reviewer’s comments very much.  

With this letter are submitting a revised version of our manuscript together with a point-to-point response to the editor and reviewers’ comments on our manuscript. The scientific language pattern in our revised manuscript has been polished and reviewed by someone that is fluent in English. We hope the revised paper will meet your expectation. All authors have read this paper and approved the resubmission.

Best wishes

Sincerely your

Lizhong Wu

Comments and Suggestions for Authors:

The Au. describe in a small cohort of patients different stages of CE in the lungs with the aim to translate the stadiation of liver cysts to those in the lungs.

1 The Au. should read this article before re-submitting the manuscript: Plos NTD, 2012, Stojkovic et al. Diagnosis and staging of cystic echinococcosis: how do CT and MRI perform in comparison to ultrasound?

Response:Thanks for your suggestion. We have read and cited this article. We are well aware that the small sample size is the shortcoming of this study. Our findings requires verification through future prospective studies with larger sample sizes.

2 The sentence about serology is not correct, in many cases serology may help in the diagnosis, the Au. do not cite any paper to support their position. Any way, a serological check of the evaluated patients would have improved the quality of the study.

Response: Blood routine and C-reactive protein is a routine examination in our hospital. So we include these results in our manuscript. In addition, tt was reported that elevated leukocytes are more common in cases with hydatid cyst rupture [1]. In the study from Dulger et al. it showed that the main laboratory characteristics of alveolar echinococcosis included mild eosinophilic leukocytosis with hypergammaglobulinemia and elevated C-reactive protein levels [2]. We have cited these papers in our revised manuscript.

Reference

[1] Ozyurtkan, M.O.; Balci, A.E. Surgical treatment of intrathoracic hydatid disease: a 5-year experience in an endemic region. Surgery today 2010, 40, 31-37.

[2] Dulger AC , Esen R , Begenik H , et al. Alveolar echinococcosis of the liver: A single center experience[J]. Polskie Archiwum Medycyny Wewntrznej, 2012, 122(4):133-138.

3 The Au. cite the WHO-IGE, but they are not aware of the Expert consensus which clearly states that all pulmonary cysts should be operated. According to WHO-ICE The stage CE3 is divided in C3a and C3b, but these stages are not considered by the Au.

Response:Thanks for pointing out. We have added Expert consensus about pulmonary cysts  in the introduction section.The added content were as follow: “The Expert consensus clearly states that all pulmonary cysts should be operated [9]”.

Meanwhile, as you point out, the stage CE3 is divided into CE3a and CE3b according to WHO-ICE. We added this missing information of CE3a and CE3b in our revised manuscript.

Reference

[9]Brunetti, E., Kern, P., Dominique Angèle Vuitton. Expert consensus for the diagnosis and treatment of cystic and alveolar echinococcosis in humans. Acta Tropica, 2010, 114(1):1-16.

4 An informtion which might be added is the presence or not of cysts in other organs, different from the lungs.

Response:Table 1 provided the information about cysts in other organs.Ten case in this study combined with liver echinococcosis.

Reviewer 2 Report

General: This is a very interesting study. The reviewer agrees with the authors that an imaging based standardized classification for pulmonary cystic echinococcosis derived from CT is missing, especially taking into account that for the lung is the most suitable organ to be diagnosed by CT imaging.

Abstract: okay.

Keywords: okay, would suggest to change “Image classification” into CT-image derived classification”

Introduction: okay. Well written.

Materials and Methods:

1) well written

Results

Nicely presented. Especially the examples for the different classifications

Some small revisions:

Table 1 and table 3: would also add to the mean ± standard deviation also the median

Figure 2:

there are just some spelling mistakes

-> G Axial CT image shows a “floating-lotus” shape -> please change the capital letter G into -> g, it also has to be written in a bold letter and put it between round brackets

-> f Axial CT image shows a “ribbon” shape -> change f into bold written letter and put it between round brackets

Dicussion:

Well written. The results are summed up with good structure. Even though the research cohort is a small one the authors could point our important facts for a reasonable classification system.

Author Response

Dear editor,

Thank you very much for your E-mail regarding our manuscript entitled “Application of multi-slice computed tomography for the preoperative diagnosis and classification of pulmonary cystic echinococcosis”. First of all, we thank editor and reviewers for constructive and helpful comments on our manuscript. We also appreciate the reviewer’s comments very much.  

With this letter are submitting a revised version of our manuscript together with a point-to-point response to the editor and reviewers’ comments on our manuscript. The scientific language pattern in our revised manuscript has been polished and reviewed by someone that is fluent in English. We hope the revised paper will meet your expectation. All authors have read this paper and approved the resubmission.

Best wishes

Sincerely your

Lizhong Wu

Comments and Suggestions for Authors:

General: This is a very interesting study. The reviewer agrees with the authors that an imaging based standardized classification for pulmonary cystic echinococcosis derived from CT is missing, especially taking into account that for the lung is the most suitable organ to be diagnosed by CT imaging.

Abstract: okay.

Keywords: okay, would suggest to change “Image classification” into CT-image derived classification”

 Response: We have change Image classification” into CT-image derived classification” in our revised manuscript.

Introduction: okay. Well written.

Materials and Methods:

1) well written

Results

Nicely presented. Especially the examples for the different classifications

Some small revisions:

Table 1 and table 3: would also add to the mean ± standard deviation also the median

Response: We have added the mean ± standard deviation and the median for the continuous data in the table 1 and table 3 in our revised manuscript.

Figure 2: there are just some spelling mistakes: -> G Axial CT image shows a “floating-lotus” shape -> please change the capital letter G into -> g, it also has to be written in a bold letter and put it between round brackets. -> f Axial CT image shows a “ribbon” shape -> change f into bold written letter and put it between round brackets.

Response: We have revised these spelling mistakes in our revised manuscript.

Dicussion:

Well written. The results are summed up with good structure. Even though the research cohort is a small one the authors could point our important facts for a reasonable classification system.

Response: Thanks for your comment.

Reviewer 3 Report

Wu et al. report on CT-based classification of cystic echinococcosis of the lungs. The work is interesting and worth presenting. However, I have some recommendations.

Major points:

  1. The main finding of the study (as shown in table 4) is the high reliability of CT scanning regarding a classification and staging of echinococcosis cysts in the lungs in line with the WHO staging for the liver. The authors convincingly show that CT scanning can reliably ensure such a staging. However, they do not convincingly show, why such a staging should be done in the lungs. In the liver, certain WHO stages have distinct therapeutic consequences, including medical treatment, surgery, PAIR, etc. As described by the authors, surgery will be the standard treatment in the lungs anyway. So, I encourage the authors to explain why and in how far a WHO-staging of cysts in the lungs will really help the surgeon regarding therapeutic decision making. The authors should use both literature recommendations and their own experience based on the therapeutic outcomes of the patients who are described in this study.

So, do the authors recommend applying the WHO classification for cysts in the lungs? If yes, which is the practical purpose of such a recommendation by means of affecting the surgeon’s decision?

  1. In the second paragraph of the discussion, the authors report that CT scanning had initially been performed for the diagnosis of upper respiratory tract infections or similar reasons. However, parallel infections might also have influenced measured blood parameters like leucocyte count or CRP. So, I am a little uncertain regarding the validity of correlating these blood parameters with stages of the cysts. The authors should discuss this hypothetical bias in the discussion chapter.

Minor points:

  1. Table 1, Grammar: I assume, “Combined liver echinococcosis” means “Combined with liver echinococcosis”, doesn’t it?
  2. Table 1, blood routine: Was blood routine performed for all patients in the same way? If not, denominators should be given.
  3. Sub-heading 2.2. is missing in the results chapter. Anyway, I recommend including more sub-headings.
  4. Results chapter: Hypothetical assumptions like “and is presumably related to secondary infection of the CE3 lesion through cyst wall rupture” are more appropriate for the discussion than for the results.
  5. Results chapter. The sentence: “There were no cases of CL lesions, which have thin cyst wall, and unclear development on CT, contained fluid similar to water.” is difficult to understand and should be reworded.
  6. Discussion: If Latin genus names like “Echinococcus” are used, they should be printed in italics.
  7. Discussion, first paragraph: The discussion should closely focus on the results. Propedeutics like the repetition of the reproductive cycle of Echinococcus spp. in the first paragraph of the discussion are superfluous and unnecessarily increase the size of the paper.
  8. Discussion, second paragraph: Indeed, eosinophils can be suggestive of parasitic infections, but they are definitely not ONLY suggestive of parasitic infections as claimed by the authors.
  9. Figure 4 and the associated contents should be shifted from the discussion chapter to the results chapter.

Author Response

Dear editor,

Thank you very much for your E-mail regarding our manuscript entitled “Application of multi-slice computed tomography for the preoperative diagnosis and classification of pulmonary cystic echinococcosis”. First of all, we thank editor and reviewers for constructive and helpful comments on our manuscript. We also appreciate the reviewer’s comments very much.  

With this letter are submitting a revised version of our manuscript together with a point-to-point response to the editor and reviewers’ comments on our manuscript. The scientific language pattern in our revised manuscript has been polished and reviewed by someone that is fluent in English. We hope the revised paper will meet your expectation. All authors have read this paper and approved the resubmission.

Best wishes

Sincerely your

Lizhong Wu

Comments and Suggestions for Authors:

Wu et al. report on CT-based classification of cystic echinococcosis of the lungs. The work is interesting and worth presenting. However, I have some recommendations.

Major points:

  1. The main finding of the study (as shown in table 4) is the high reliability of CT scanning regarding a classification and staging of echinococcosis cysts in the lungs in line with the WHO staging for the liver. The authors convincingly show that CT scanning can reliably ensure such a staging. However, they do not convincingly show, why such a staging should be done in the lungs. In the liver, certain WHO stages have distinct therapeutic consequences, including medical treatment, surgery, PAIR, etc. As described by the authors, surgery will be the standard treatment in the lungs anyway. So, I encourage the authors to explain why and in how far a WHO-staging of cysts in the lungs will really help the surgeon regarding therapeutic decision making. The authors should use both literature recommendations and their own experience based on the therapeutic outcomes of the patients who are described in this study.

Response: Thanks for your suggestion. Surgery for hydatid disease includes hydatid cyst resection, segmental wedge resection, lobectomy, internal capsule removal, and cyst closure (Expert consensus for the diagnosis and treatment of cystic and alveolar echinococcosis in humans).The specific surgical plan is mainly based on the location, size, type, and complications of the lesion shown on the preoperative CT(2).The principle of surgery is to remove the entire lung hydatid cyst lesion as completely as possible. At the same time, the functional lung tissue is preserved as much as possible, and bronchial asphyxia caused by cyst fluid overflow or severe systemic allergic reactions must be absolutely avoided during the operation (3, 11). According to the treatment results of our study, we recommend that patients with CE1, CE2, CE4, and CE5 type mainly undergo hydatid cyst resection and patients with CE3 type undergo pulmonary segmental or lobectomy according to the scope of imaging and whether there are complications. Patients with CE3 who have difficulty in resection are advised to undergo internal capsule removal. We have added this recommendations in the discussion section(paragraph 5) of our revised manuscript.

Reference:

1 Brunetti, E., Kern, P., Dominique Angèle Vuitton. Expert consensus for the diagnosis and treatment of cystic and alveolar echinococcosis in humans. Acta Tropica, 2010, 114(1):1-16.

2 Gottstein, B.; Reichen, J. Hydatid lung disease (echinococcosis/hydatidosis). Clinics in chest medicine 2002, 23, 397-408, ix.

3 Shehatha, J.; Alizzi, A.; Alward, M.; Konstantinov, I. Thoracic hydatid disease; a review of 763 cases. Heart, lung & circulation 2008, 17, 502-504.

4 Engström, E.L.S.; Salih, G.N.; Wiese, L. Seronegative, complicated hydatid cyst of the lung: A case report. Respiratory medicine case reports 2017, 21, 96-98.

So, do the authors recommend applying the WHO classification for cysts in the lungs? If yes, which is the practical purpose of such a recommendation by means of affecting the surgeon’s decision?

Response: We recommend applying the WHO classification for cysts in the lungs. The specific surgical plan is mainly based on the location, size, type, and complications of the lesion shown on the preoperative CT. These have been addressed in the discussion section.

  1. In the second paragraph of the discussion, the authors report that CT scanning had initially been performed for the diagnosis of upper respiratory tract infections or similar reasons. However, parallel infections might also have influenced measured blood parameters like leucocyte count or CRP. So, I am a little uncertain regarding the validity of correlating these blood parameters with stages of the cysts. The authors should discuss this hypothetical bias in the discussion chapter.

Response: Thanks for pointing out. We have discussed this hypothetical bias in the discussion chapter in our revised manuscript. The added content were as follow:“Parallel infections might have influenced measured blood parameters like leucocyte count or C-reactive protein.In this study, we did not consider this bias and therefore correlation between increased leukocytes and stages of the cysts still need be furtherly verified.”

Minor points:

  1. Table 1, Grammar: I assume, “Combined liver echinococcosis” means “Combined with liver echinococcosis”, doesn’t it?

   Response: Thanks for pointing out. We have revised “Combined liver echinococcosis” as “Combined with liver echinococcosis” in our revised manuscript.

  1. Table 1, blood routine: Was blood routine performed for all patients in the same way? If not, denominators should be given.

      Response: Blood routine is a routine examination in our hospital. Blood routine was performed for all patients in the same way in this study.

  1. Sub-heading 2.2. is missing in the results chapter. Anyway, I recommend including more sub-headings.

Response:  We have added two sub-headings about the 2.2 section in our revised manuscript.

  1. Results chapter: Hypothetical assumptions like “and is presumably related to secondary infection of the CE3 lesion through cyst wall rupture” are more appropriate for the discussion than for the results.

Response: Thanks for your suggestion. We have deleted these hypothetical assumptions in result section of our revised manuscript.

  1. Results chapter. The sentence: “There were no cases of CL lesions, which have thin cyst wall, and unclear development on CT, contained fluid similar to water.” is difficult to understand and should be reworded.

   Response: We have reworded this sentence as “There were no cases of CL lesions found. These lesions demonstrate unclear development on CT and have a thin cyst wall and contain fluid similar to water” in our revised manuscript.

  1. Discussion: If Latin genus names like “Echinococcus” are used, they should be printed in italics.

   Response: Thanks for pointing out. The Echinococcus in our revised manuscript have 

   been printed in italics.

  1. Discussion, first paragraph: The discussion should closely focus on the results. Propedeutics like the repetition of the reproductive cycle of Echinococcus spp. in the first paragraph of the discussion are superfluous and unnecessarily increase the size of the paper.

   Response: Thanks for our comment. We have decreased the size of first paragraph.

  1. Discussion, second paragraph: Indeed, eosinophils can be suggestive of parasitic infections, but they are definitely not ONLY suggestive of parasitic infections as claimed by the authors.

Response: Thanks for pointing out. We have changed the “Eosinophils are only suggestive of parasitic infection” into  “Eosinophils are suggestive of parasitic infection”.

  1. Figure 4 and the associated contents should be shifted from the discussion chapter to the results chapter.

Response: Thanks for your suggestion.The Figure 4 have been shifted from the discussion chapter to the results chapter in our revised manuscript.

Round 2

Reviewer 1 Report

The Au. have addressed all the criticisms raised by me.